# The RARγ Oncogene: An Achilles Heel for Some Cancers

**DOI:** 10.3390/ijms22073632

**Published:** 2021-03-31

**Authors:** Geoffrey Brown, Kevin Petrie

**Affiliations:** 1Institute of Clinical Sciences, School of Biomedical Sciences, College of Medical and Dental Sciences, University of Birmingham, Edgbaston, Birmingham B152TT, UK; 2School of Medicine, Faculty of Health Sciences and Wellbeing, University of Sunderland, Sunderland SR13SD, UK; kevin.petrie-1@sunderland.ac.uk

**Keywords:** retinoic acid receptor γ, oncogenes, leukemia, carcinoma, stem cells

## Abstract

Cancer “stem cells” (CSCs) sustain the hierarchies of dividing cells that characterize cancer. The main causes of cancer-related mortality are metastatic disease and relapse, both of which originate primarily from CSCs, so their eradication may provide a bona fide curative strategy, though there maybe also the need to kill the bulk cancer cells. While classic anti-cancer chemotherapy is effective against the dividing progeny of CSCs, non-dividing or quiescent CSCs are often spared. Improved anti-cancer therapies therefore require approaches that target non-dividing CSCs, which must be underpinned by a better understanding of factors that permit these cells to maintain a stem cell-like state. During hematopoiesis, retinoic acid receptor (RAR) γ is selectively expressed by stem cells and their immediate progeny. It is overexpressed in, and is an oncogene for, many cancers including colorectal, renal and hepatocellular carcinoma, cholangiocarcinomas and some cases of acute myeloid leukemia that harbor RARγ fusion proteins. In vitro studies suggest that RARγ-selective and pan-RAR antagonists provoke the death of CSCs by necroptosis and point to antagonism of RARγ as a potential strategy to treat metastatic disease and relapse, and perhaps provide a cure for some cancers.

## 1. Introduction

The stem cell theory of cancer states that most, if not all, cancers arise from a tissue-specific stem cell [1]. When an oncogenic insult modifies one of the stem or progenitor cells that give rise to various types of mature tissue cells, it converts it into a cancer-initiating cell. This cell is the origin of cancer, but more changes are needed before it, or an offspring of the cancer-initiating cell, becomes a cancer stem cell (CSC) [2]. This cell sustains the hierarchy of dividing cells that characterizes cancer.

Classic anti-cancer chemotherapy is effective against the dividing progeny of CSCs, but spares non-dividing CSCs. As early as 1999, a rare and quiescent subpopulation of primitive leukemia cells was isolated from samples from 6 patients with chronic-phase chronic myeloid leukemia (CML). These leukemia stem cells (LSCs) were insensitive to even high doses of cytotoxic agents targeted against the cell cycle [3]. CML LSCs have also been found to be resistant in vitro to the tyrosine kinase inhibitors, such as imatinib, that are used to treat CML and many different types of cancer [4,5]. LSCs are envisaged to be insensitive to imatinib in vivo. Curing CML patients is also very challenging because LSCs that are resistant to treatment can lead to disease relapse following remission, and even after allogeneic transplantation [6]. Accordingly, relapse is the main cause of leukemia-related mortality.

CSCs are also largely the cause of carcinoma metastases. Whilst cancer treatment has made great strides in the last 40 years, the treatment of relapsed and metastatic disease has not advanced significantly. A view of clinicians is that once a cancer has progressed to metastatic disease it is often beyond successful treatment by conventional chemotherapeutics. For example, stage 4 metastatic colorectal cancer (CRC) is considered terminal, with the 5-year survival being 5% for men and 10% for women. Finding a way to eradicate CSCs will undoubtable reduce cancer mortality in today’s world. The era of the COVID-19 pandemic has thrown this effort into sharp relief, with severe disruption in cancer screening, diagnosis and elective surgery. This scenario is anticipated to lead to an increase in untreatable cancers and premature deaths. The development of new therapies that are capable of targeting CSCs to avoid and treat aggressive and metastatic disease is therefore a priority. There may also be the need to kill the bulk of the cancer cells that are the progeny of CSCs and that kill the patient. Therapies targeted to CSCs may also kill non-CSC cells or be effective also against the latter cells due to the exhaustion of CSCs. Alternatively, anti-CSCs agents will be used in conjunction with chemotherapeutics to deal with the cancer burden. To achieve the targeting of CSCs, identification of oncogenic events within CSCs is critical. Here, we examine evidence that supports the view that RARγ is an Achilles heel for some cancers.

## 2. The Concept of the Cancer Stem Cell

Evidence in support of the concept of CSCs is underpinned by investigative studies, in particular in acute leukemias that led to the characterization of LSCs. The first step towards the identification of LSCs was the demonstration that the cells from patients with acute myeloid leukemia (AML) are a hierarchy of cancer cells that parallel the well-described development of cell lineages during normal hematopoiesis [7,8]. Cells from AML patients formed colonies in vitro in the colony forming assay that is used to describe the various developmental stages of normal bone marrow cells [9]. The identity of the cells giving rise to the AML colony-forming units (AML-CFU) was determined by using monoclonal antibodies to cell surface antigens and complement to lyse cells from a number of patients followed by the use of the colony-forming assay [10]. AML-CFU were phenotypically distinct from patients’ bulk blast cell populations and classified as similar to (i) primitive and pluripotent normal colony forming units from normal bone marrow, (ii) early granulocyte/macrophage lineage-committed colony-forming units, and (iii) late granulocyte/macrophage lineage-committed colony-forming units.

When assayed using 3D and co-culture techniques, human hematopoietic stem cells (HSC) and primitive progenitors (Long-Term Culture Initiating cells, LTC-IC), display similar phenotypes, which are distinct from normal colony-forming progenitors [11]. Even so, HSC are truly defined by their ability to reconstitute hematopoiesis in the severe combined immunodeficient (SCID) mouse [12] and subsequent models [13]. It was thus predicted that AML “stem cells” would share these characteristics and have the ability to engraft upon transplantation into SCID mice. This was indeed the case, and leukemia engraftment in SCID mice was observed for cells from most AML subtypes, with the exception of acute promyelocytic leukemia (APL) [14]. The frequency of AML cells that engrafted was very low, at around 1 per 10^6^ cells, and much lower than the frequency of AML-CFU within patients’ cells, at 1 in 100. Moreover, leukemias could also only be engrafted in mice using primitive CD34+CD38- cells, whereas AML-CFU were most prevalent in the CD34+CD38+ cell population. The primitive leukemia cells that could be engrafted were termed SCID leukemia-initiating cells and, in essence, are AML “stem cells”.

While the characterization of cells within solid tumors that possess a stem cell phenotype is less advanced than for the leukemias, CSCs for various carcinomas have been described (reviewed in [15]), including those for colorectal cancer (CRC) [16], breast cancer (BC) [17], brain cancer [18], head and neck squamous cell carcinoma [19], melanoma [20,21] and pancreatic cancer [22]. Similar to AML, these populations of cells may engraft efficiently in mice, including by a single CSC in the case of melanoma. The proportion of CSCs within a tumor varies, ranging from very few cells to up to 25%, and this large difference probably relates to the nature of the different types of cancer. As to whether carcinoma CSCs are bona fide, gene expression profiling has shown that they express stem/progenitor associated genes. The presence and/or predominance of CSCs is important to the outcome from treatment. For example, brain CSCs are resistant to radiation in vitro and in vivo, due to activation of the DNA damage response [23]. In the case of BC, analysis of 186 genes differentially expressed by BC CSCs compared with normal breast epithelium has led to the identification of an “invasiveness” signature associated with risk of metastasis and death [24].

## 3. Targeted Therapies for Cancer

The ongoing focus in the endeavor to develop new cancer treatments is the design of drugs that are targeted to a specific type of cancer, to personalize treatment and extend the realm of conventional chemotherapeutics [25]. As to the significance of LSCs/CSCs for disease outcome, whether the new treatments that are developed adversely affect LSCs/CSCs is a critical consideration. In the case of AML, LSCs are resistant in vitro to daunorubicin [26] and to Arc-C [27]. The cell surface receptor fms-like tyrosine kinas 3 (Flt3) has been used to identify HSCs and progenitor cells [28]. Flt3 is a class III tyrosine kinase that has structural homology to the c-kit and macrophage colony-stimulating factor (CSF1) receptors. Expression in humans is restricted to CD34+ bone marrow cells which includes HSCs [29]. Recent in vivo studies have revealed an instructive role for the ligand for Flt3 whereby it drives multipotent hematopoietic progenitor cells towards the myeloid/lymphoid and away from the megakaryocyte/erythroid lineages [30]. As an approach to treating AML, Flt3 mutations, that occur in 30% of case, have been targeted by the use of small molecule tyrosine kinase inhibitors that are effective against Flt3. However, and so far, the early use of the inhibitors has met with modest success. As to the outcomes from early trials. there have been just short-lived responses of peripheral blast cells and bone marrow blast cells have responded less frequently [31]. Antibodies to Flt3 might provide another way of targeting Flt3, to inhibit signaling, to treat AML. There are now new-generation Flt3 inhibitors that are more specific, more potent and less toxic. Even so, acquired resistance to inhibitors remains a challenge and very low 5-year survival levels from first relapse of around 10% underline the urgent need for new treatments [32].

Perhaps the most successful example to date of a precision medicine approach that efficiently targets cancer stem cells is the use of all-*trans* retinoic acid (ATRA)-based therapy to treat APL [33]. APL, which is classified as M3 under the French-American-British (FAB) system, accounts for between 5–15% of total cases of AML. The most common form of APL, characterized by the t(15;17)(q24;q21) translocation leading to the creation of a fusion between the *PML* and *RARA* (retinoic acid receptor) genes. While APL is cytogenetically a less complex disease compared with non-APL AML, it is similarly defined by a hierarchy of cancer cells as evidenced by the presence of the oncogenic PML-RARα fusion protein in patients’ LSCs [34]. For APL, ATRA in combination with arsenic trioxide induces differentiation and LSC clearance, which is curative in over 95% of patients [35,36]. Therapeutic ATRA treatment (10^−6^ M) does not significantly affect normal hematopoiesis but instead targets the oncogenic PML-RARα fusion protein, promoting disassociation of transcriptional co-repressors from both PML-RARα and wild type RARα as well as their proteolytic degradation to drive differentiation and apoptosis. In vitro studies using the APL-like NB4 cell line have identified a role for aberrant histone acetylation to chromatin in the pathogenesis of this disease, which was relieved by ATRA treatment [37]. Furthermore, the combination of ATRA and arsenic trioxide induced greater demethylation of target genes in APL cells, which may be important for lasting differentiation and remission [38]. Underscoring the importance of epigenetics, recent research in non-APL AML has suggested that epigenetic reprogramming of genes involved in differentiation, proliferation and survival effectively sensitizes LSC to ATRA-based therapy [39,40]. In the case of solid tumors, such as head and neck, ATRA targets CSCs to drive apoptosis and there is an impact on tumor control when combined with radiotherapy [41].

The requirement for the development of new therapeutic approaches is equally pressing for solid tumors. For solid tumors, a general therapeutic approach is to prevent the growth of new blood vessels that feed the tumor. As mentioned above, the prognosis of stage 4 metastatic CRC is very poor and the 5-year survival for metastatic prostate carcinoma (PCa) is ~29%. Vascular endothelial growth factor (VEGF) promotes tumor angiogenesis and two different approaches to target VEGF have been investigated and evaluated in clinical trials. Firstly, direct targeting of VEGF-A by bevacizumab (Avastin), a humanized monoclonal antibody [42], and secondly, enzymatic inhibition of VEGF receptor tyrosine kinase activity using vatalanib [43]. The addition of bevacinzumab to chemotherapy and its use in other combinatorial strategies has been demonstrated to improve survival and it now a standard addition to frontline treatment of advanced CRC and represents key therapeutic approach in cancer [42]. Development of vatalanib, on the other hand, was discontinued due toxicity issues [44]. Whilst excessive abnormal angiogenesis is important to solid tumor progression, an alternative view is that promotion of vessel maturation, rather than to ablate vessels, could reduce tumor hypoxia and improve drug uptake [45].

In PCa, targeted approaches have focused on androgen-deprivation in combination with chemotherapy or radiotherapy [46,47]. Unfortunately, a highly aggressive and metastatic hormone-independent form of the disease (castration or hormone-resistant prostate cancer) develops in a significant fraction of patients receiving treatment, leading to treatment failure. Treatment options for these patients are extremely limited and even with the introduction of better therapeutics, the average survival time of patients with hormone-resistant PCa is still often less than 36 months [46,47]. Presently, investigators are looking to delineate the androgen signaling pathways in PCa to develop new treatments [48].

## 4. RARγ Is a Fundamental Control on Stem Cell Behavior

There are three retinoic acid receptor (RAR) isotypes (RARα, RARβ, RARγ) that heterodimerize with retinoid X receptors (RXRα, RXRβ, RXRγ) and bind to retinoic acid response elements (RAREs). Upon binding their physiological ligand, all-*trans* retinoic acid (ATRA), undergo coregulator exchange to activate target genes and modify cell behavior. It is well known that an appropriate level of ATRA is crucial to the conduct of normal embryonic development [49]. In this regard, the distribution and levels of ATRA in embryonic tissue is very tightly regulated [50]. Various mechanisms guard against dysregulated ATRA synthesis and metabolism [51]. ATRA also plays a crucial role in the differentiation of embryonic stem cells, with both classical genomic effects and non-genomic events, such as the activation of kinase cascades, playing a role [52]. Additionally, the RARα and RARγ regulatory networks within cells are highly extensive, as revealed by integrative genomics dissection of the differentiation of F9 embryonal stem cells [53]. Therefore, perturbations to the control of cell behavior by RARs might be expected to underlie the abnormal behavior of some cancer cells, and, as below, this is indeed the case.

RARs are key regulators of gene expression within stem cells, with the RARα and RARγ isotypes playing critical and opposing roles in hematopoiesis. RARγ is selectively expressed in hematopoietic stem cells (HSCs) and their immediate progeny (Lineage^−ve^, Sca-1^+ve^, c-Kit^+ve^ (LSK)), and its activation by ATRA is vital for the maintenance of HSC numbers. *Rarg* knockout mice display markedly reduced numbers of HSCs with a concomitant increase in more mature progenitors [54]. By contrast, RARα-mediated transcription drives neutrophil differentiation of myeloid progenitors. It is important to bear in mind that evidence strongly suggests the activities of both RARα and RARγ are regulatory, rather than obligatory, for myeloid cell development, which persists in mice in the context of knockouts of *Rara* and *Rarg*. The roles of RARα and RARγ are also linked to other controls on myeloid differentiation. The cytokine granulocyte colony-stimulating factor (G-CSF) is important to myelopoiesis and RAR-regulated cellular events interact with signaling from the G-CSF receptor (G-CSFR) [55]. Similar to RARs, G-CSFR is dispensable, and can therefore be considered regulatory. However, in mice lacking G-CSFR, cellular expansion associated with granulopoiesis fails to occur, a phenotype that is also displayed upon treatment with the pan-RAR antagonist AGN19430 [56]. We can therefore conclude that both RARα and RARγ are critical for the proper conduct of hematopoiesis, at least, whereby they promote cell differentiation and self-renewal, respectively. The crosstalk between RARs and G-CSFR is an intriguing aspect of the functionality of RARs and it is likely that other examples remain to be identified.

Findings from studies of zebrafish development further support the view that RARγ activity is important for the maintenance of stem/progenitor cell populations [57]. In zebrafish, there are two major RAR isotypes, RARα and RARγ, and treatment of zebrafish embryos four hours post-fertilization with the RARγ-selective agonist AGN205327 blocked the development and growth of tissues that are derived from cranial neural crest and primitive mesoderm. Somite formation and axial length were reduced, and the formation of cranial bones and anterior neural ganglia that arise from the cranial neural crest was disrupted. AGN205327 treatment also blocked pectoral fin outgrowth and caudal fin regeneration after transection. The lateral plate mesoderm stem/progenitor cells associated with pectoral fin tissue were intact as evidenced by the presence of pectoral fin buds and Tbx-5 positive immunostaining. Moreover, the AGN205327-dependent block of pectoral fin development was reversed by co-treatment with an RARγ-selective antagonist AGN205728, or by RARγ agonist washout. Thus, RARγ supports a stem cell phenotype by actively blocking development and differentiation.

Various findings from studies of mouse embryonal stem cells point towards RARγ influencing the lineage fate of stem cells. F9 embryonal carcinoma cells lacking both alleles of *Rarg* display diminished expression of genes that are central to cell fate determination such as Hoxa1 [58], which patterns the hindbrain and craniofacial features during embryonic development. Within F9 embryonal carcinoma cells, Rar/Rxr dimers specifically interact with genomic regions that are associated with pluripotency, as characterised by binding of pluripotency-associated transcription factors [59]. Chromatin structure is intimately connected with cell pluripotency and findings from studies of mouse *Rarg* knockout embryonic stem cells have revealed that active Rarγ plays a role in ATRA-induced chromatin remodelling [60]. In embryonic stem cells, Rarγ signalling is essential for the expression of most of the ATRA-regulated transcriptome, including the transcription factor Meis1 (myeloid ecotropic viral integration site-1). Meis1 expression was observed to be increased when wild type (WT), but not *Rarg* knockout (*Rarg*^−/−^), cells were treated with ATRA. In WT, but not in *Rarg*^−/−^ cells, there was also a rapid increase in the transcriptionally permissive epigenetic markers, histone H3 acetylated Lys9/Lys14 (K9^Ac^/K14^Ac^), at sites downstream and proximal to transcription start site of *Meis1*. In other words, loss of Rarγ prevented ATRA-driven epigenetic changes at the *Meis1* gene. The means by which Rarγ can provoke changes to the epigenetic signature remain unclear because RAREs are not present, nor was Rarγ binding detected in either the downstream or proximal regions of the *Meis1* promoter.

The above evidence strongly suggests that RARγ is a key regulator of the behaviour of stem cells. There are two possibilities regarding how RARγ might have an oncogenic role within stem cells and/or their progeny. First, overexpression might have a direct action to dysregulate the expression of key target genes and, in turn, change cell behaviour. Alternatively, it is important to bear in mind that RARγ provokes epigenetic changes within stem cells and that epigenetic signatures are increasingly considered as important to cancer, cell responsiveness to chemotherapeutics, and targeting therapeutic interventions. Some oncogenes influence the fate of stem cells: *LMO2*, *BCR-ABLp190*, and *BCR-ABLp210* are specifically associated with and drive human T acute lymphoblastic, B acute lymphoblastic leukemia, and chronic myeloid leukemia (CML), respectively. When the expression of each of these oncogenes was restricted to HSCs in transgenic mice, each oncogene led to the respective human-like lineage-restricted leukemia [61,62,63]. It seems, therefore, that that each oncogene had set the choice of the targeted leukemia stem cell (LSCs), as well as their progeny, towards a distinct developmental pathway. The different global patterns of DNA methylation at promoters for LSCs from each of the mice led the investigators to conclude that the tumor stem cell priming, to fix the lineage fate of the leukemia cells, was primarily due to epigenetic control. Similarly, different mutant RUNX1 oncoproteins, which are drivers of some case of acute myeloid leukemia, were also observed to program distinct HSC lineage trajectories, skewing cells towards either a B cell or megakaryocyte/erythroid identity. The different mutant proteins were also observed to alter lineage-specific chromatin priming [64]. Another aspect to the importance of epigenetics in leukemia is the role of DNA methytransferases alongside ATRA in the regulation of expression *Hox* genes, which are critical for the proper maintenance of HSC, as well as proliferation and survival of LSC [65].

## 5. RARγ Is an Oncogene for Some Cancers

Recent discoveries are that RARγ is a molecular driver of the cancers summarized in Table 1, and overexpression occurs other than in AML whereby RARγ fusion proteins are seen in a rare subtype of AML.

As outlined above, the roles of *RARA*-associated gene rearrangements in the pathophysiology of APL have been well-characterized and patients are cured by the use of ATRA with arsenic trioxide [36]. Recently, cases of APL that are characterized by rearrangements of the *RARG* gene rather than the *RARA* gene have been identified [66]. Here, among nine patients, fusions were observed between the *RARG* gene and the genes for *NUP98* (3 patients), *CPSF6* (4 patients), *NPM1* (1 patient) and *PML* (1 patient). The mechanisms underlying the transformations by the fusion proteins remain to be established. The NUP98 (nucleoporin) N-terminal region acts as a transcription activator, CPSF6 is subunit of cleavage factor 1 (an RNA binding protein), NPM1 (nucleophosmin) is a nucleolar protein that participates in chromatin modelling, and PML is a tumor suppressor protein. With the exception of the patient harboring the *PML-RARG* translocation, all the other cases with *RARG* gene rearrangements failed to respond to ATRA-based therapy. For patients with *RARG* gene rearrangements there is therefore the need for new treatment strategies.

The potential impact of RARγ on disease progression in AML was brought to attention by a case of AML that was characterized by a novel t(4;15) (q31;q22) translocation involving the *TMEM154* and *RASGRF1* genes. Here, a 30-year-old woman with relapsed AML was treated for eight days with ATRA as part of investigational study but died with rapid disease progression. Primary AML cells taken from the patient proliferated rapidly when treated ex vivo with ATRA, as well as isotype-selective agonists of RARα and RARγ, and there was an increase in the levels of nuclear RARγ upon ATRA treatment. Whilst requiring further investigation, the basis of the growth-stimulatory response to ATRA could be associated with the increase in the level of RARγ within the nucleus following ATRA treatment [67].

RARγ is frequently overexpressed in human CRC [68], cholangiocarcinoma (CCA) [69] and hepatocellular carcinoma (HCC tissues) [70]. 73% of CRCs overexpress *RARG* mRNA and protein compared with 20% for adjacent non-tumor colorectal tissue (20%). CRC cell lines, such as HT29, HCT116, RK0 and SW480, also overexpress RARγ and knockdown of *RARG* enhanced the sensitivity of CRC cells lines to the chemotherapeutic drugs 5-fluorouracil, oxaliplatin and vincristine sulphate. Additionally, the level of the multidrug resistance protein 1 (MDR1) was diminished in knockdown cells alongside suppression of the Wnt/β-catenin pathway [68]. Chemo-resistant bile duct carcinoma CCA is the second most common hepatic malignancy and RARγ overexpression is associated with a poor prognosis and resistance to 5-flurouracil [69]. In agreement with these findings, RARγ interacted with β-catenin in CCA cells, leading to its nuclear translocation. Furthermore, shRNA-mediated knockdown of RARγ reduced proliferation, migration and invasion in vitro, and xenograft engraftment in vivo.

In HCC, overexpression of RARγ is found in the majority of primary tissues as well as HCC cell lines. In HepG2 HCC cells, overexpression RARγ promoted colony formation and xenograft engraftment [70]. Further evidence for a role for RARγ in HCC is the finding that in the SMMC-7721 cell line, expression of microRNA-30a-5p (which has been shown to downregulate *RARG*, manuscript in preparation) suppressed the proliferation and invasion [71]. Additionally, in clear cell renal cell carcinoma (ccRCC), around 50% of tissues overexpress RARγ [72], which may contribute to disease progression.

While both PCa cells and normal prostate epithelium express RARα and RARγ together, in the case of PCa, the tumor cells appear to be dependent on active RARγ for their proliferation and survival. The level of ATRA in primary PCa tissues is close to the limit of detection (at approximately 1 ng/gram tissue), whereas in adjacent normal tissues and benign prostate hyperplasia levels up to 8 times greater are found [73]. Results from LNCaP PCa cells suggest this scenario will confer dependency on the action of RARγ. This is because a 0.24 nM concentration of ATRA (equivalent to 1 ng/gram tissue) is sufficient to transactivate RARγ but not RARα, which requires a higher concentration of 19.3 nM ATRA (Figure 1).

## 6. Antagonising RARγ Kills Cancer Stem Cells

The action of antagonists of RAR on multiple cancer types provides strong support to the notion that RARγ possesses oncogenic potential. Early studies investigated in PCa cell lines and primary cells the activity of the pan-RAR antagonist AGN194310, which has high affinities for RARα, β and γ (Table 2) but not retinoid X receptors. At an IC_50_ value of 16 nM, which is close to the K_d_ values for RARα, β and γ, AGN194310 treatment led to growth arrest and cell death of DU145, LNCaP and PC3 PCa cell lines cultured in flasks (which contain a high proportion of non-CSC-like cells) [74]. AGN194310 treatment also ablated colony formation (which is dependent on the activities of CSC-like cells) by PCa cell lines as well as the breast cancer cell lines MCF-7 and MDA-MB-213 (unpublished data). Similar results were observed for primary PCa cells but not normal prostate epithelial cells, which were 50% less sensitive (discussed later) [74]. Thus, for both cell lines and primary cells, AGN194310 demonstrated efficacy against both CSC-like and non-CSC-like cells.

The importance of RARγ activation for the survival of PCa CSC-like cells may explain why patient-derived PCa cells were more sensitive to AGN194310. This is because the bovine serum albumin (BSA)-supplemented medium used to culture patient-derived PCa cells and normal prostate epithelium when testing the pan-RAR antagonist AGN194310 would likely contain a concentration of ATRA sufficient to activate RARγ but not RARα. This notion was further strengthened by experiments using highly selective antagonists for RARα and RARγ: AGN195183 and AGN205728, respectively (Table 2). Consistent with idea that RARα does not play a functional role in the maintenance of a stem cell phenotype in PCa cells, its antagonism had no effect. By contrast, when used at concentration of 5 nM (IC_50_) (which is close to its K_d_ value), treatment with the RARγ-selective antagonist AGN205728 led to growth arrest and cell death in cell lines (which was synergistic in combination with docetaxel). In agreement with the data from AGN194310, AGN205728 also inhibited the colony-forming potential of PCa [75] and breast cancer cell lines (unpublished data), again suggesting that both CSC-like and non-CSC-like cells were targeted. Confirming a survival-supportive role for RARγ within PCa cells, the RARγ-selective agonist AGN205327, and also 0.1 nM ATRA (a concentration that activates only the RARγ isotype), both increased PCa colony formation. Non-tumorigenic RWPE-1 cells did not exhibit increased colony formation when treated in the same way. These findings, together with the identification of miR-30a-5p as a tumor suppressor in PCa [76], strongly support to the view that RARγ does indeed represent a potential Achilles heel for PCa.

Tumors of the central and peripheral nervous system are the most common cause of cancer mortality in children and there is an urgent need to improve treatment options. Primitive neuroectodermal tumors (PNETs) include neuroblastoma, Ewing’s sarcoma, retinoblastoma, medulloblastoma and supratentorial primitive neuroectodermal tumors (stPNETs). Primary cells biopsied from these tumors will harbor specific genetic abnormalities (Figure 2a) but with nevertheless similar phenotypes. The culture of primary PNET cells in Neurocult (a specialized serum-free neural stem cell medium, STEMCELL Technologies) supplemented with 20 ng/mL epidermal growth factor, leads to the formation of neurosphere-like structures that generate cells that migrate and differentiate (Figure 2b). The cells that give rise to the immature and differentiating cells within neurospheres are CSCs. These cells and their progeny were ablated by the pan-antagonist AGN194310 (Figure 2c). It was substantially more effective against two pediatric PNETs and a pediatric astrocytoma than ATRA and the RARα-selective agonist AGN195183. Human fibroblasts and blood mononuclear cells were insensitive to AGN194310.

## 7. The Oncogenic Action of RARγ

Cell death provoked by the RAR pan-antagonist AGN194310 and the RARγ-selective antagonist AGN205728 was found to be mitochondria-dependent and caspase-independent [74,75]. This process, seen also for retinoid-deprived Jurkat T-cell leukemia cells, is termed necroptosis and it occurs via poly-(ADP-ribose) polymerase PARP-1 activation [77]. The PARP-1 inhibitor 1,5-dihydroisoquinoline blocked the effects of AGN205728 (unpublished data). PARP-1 ribosylates proteins to change their function leading to mitochondrial release of ATP, NAD^+^ and caspase-independent nucleases that fragment DNA. Necroptosis is considered to be a “fail-safe” cell death pathway for apoptosis-resistant cells [78], and these data support the notion that active RARγ contributes to blocking this process, potentially representing a significant aspect of its oncogenic action. At least 16 BCL-2 family members control cell survival by mediating apoptosis and/or necroptosis, and it is plausible that RARγ may play a role in controlling their expression. Indeed, ATRA has been shown to up-regulate expression of the pro-survival BCL2 family members MCL-1 and BCL-W [79].

Although a clear understanding of the role of RARγ in necroptosis remains to be established, the function of a cytosolic form of RARγ seems to be important for TNF-induced cell death in HT-29 colorectal adenocarcinoma cells [80]. When cellular inhibitor of apoptosis (cIAP) activity was blocked, RARγ controlled receptor-initiating protein kinase 1 (RIP1)-initiated cell death by mediating dissociation of RIP1 from TNFR1. Moreover, recent studies using *Rarg* knockout mice and primary squamous cell carcinoma cells have shown that loss of RARγ abolishes DNA damage-induced necroptosis [81]. Here, in the absence of RAR, the death complex known as the Ripoptosome (RIPK1)/RIPK3) failed to form. It is therefore reasonable to speculate that whilst agonizing RARγ leads to its accumulation in the nucleus [67], its antagonism might lead to retention or accumulation in the cytoplasm, potentially driving necroptosis.

## 8. Concluding Remarks

The studies described above provide evidence that RARγ functions as a pivotal oncogene in the progression of some cases of AML, CRC, CCA, HCC and renal cell cancer. There is also evidence to support an oncogenic role in PCa, PNET and ACT. Antagonism of RARγ is highly effective in vitro in ablating PCA and BC CSC-like cells, as well as primary CSCs in PNET and ACT. The finding that RARγ-selective antagonists provoked cell death of CSCs via necroptosis represents a novel treatment strategy. The potential of this approach is further enhanced by the observation that normal cells seem to be less susceptible to necroptosis following antagonism of RARγ, which might have the benefit of provoking fewer side-effects. While pan-RAR and RARγ-selective antagonism show great promise, the results thus far have been obtained in vitro and it is necessary to assess their efficacy in vivo. Finally, further work is also required to fully ascertain the mechanisms by which RARγ maintains a stem-cell phenotype. Addressing these questions should provide answers regarding the importance of RARγ in cancer initiation and/or disease progression.

## Figures and Tables

**Figure 1 ijms-22-03632-f001:**
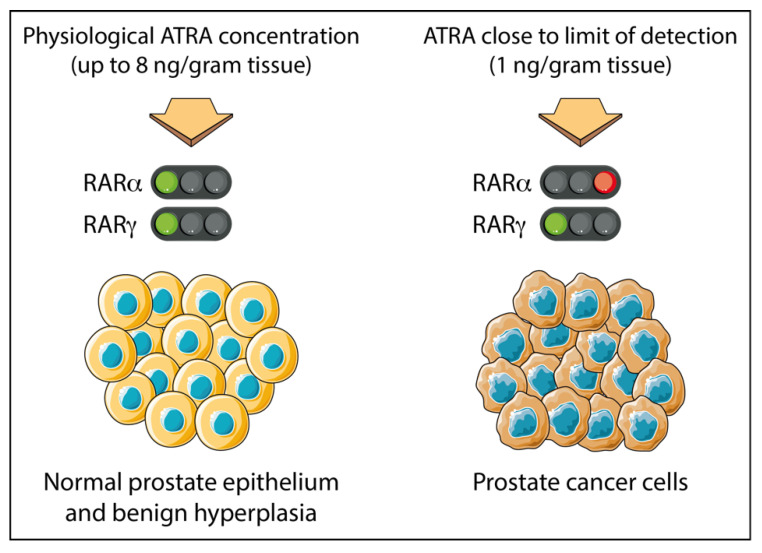
Prostate cancer tissue is adapted to grow in sub-nanomolar concentrations of ATRA. Adjacent normal cells and benign prostate hyperplasia cells see ~8 times more ATRA than level within patients’ PCa tissue which is close to the level of detection at ~1 ng/g tissue [28]. For patients’ PCa tissue, there is sufficient to transactivate RARγ, which requires just 0.24 nM ATRA (see Table 2). RARα requires 19.3 nM ATRA for transactivation (see Table 2) and therefore is unlikely to be active within patients’ PCa tissue. This figure was created using Servier Medical Art templates, which are licensed under a Creative Commons Attribution 3.0 Unported License; https://smart.servier.com, accessed on 28 February 2021.

**Figure 2 ijms-22-03632-f002:**
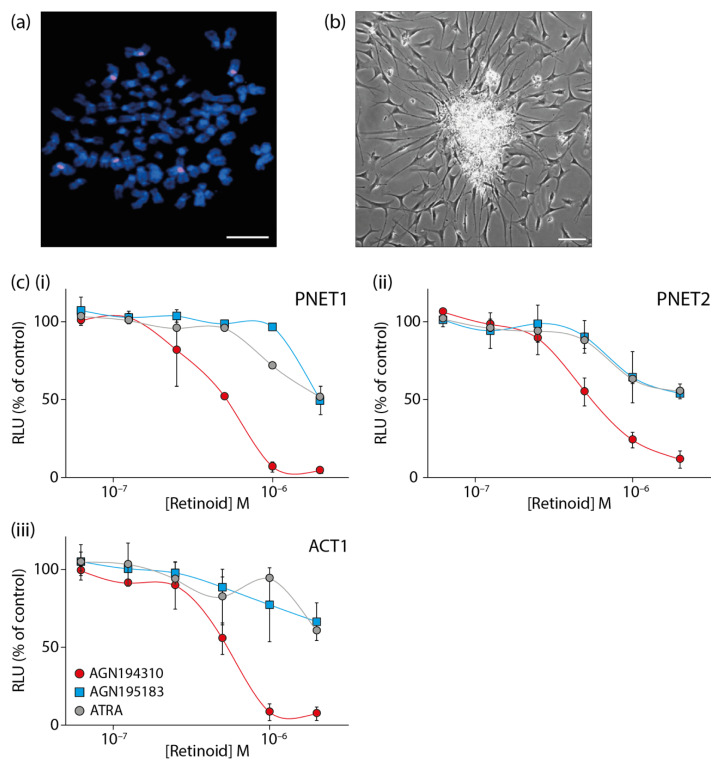
The pan-RAR antagonist AGN194310 ablates primary cells grown from pediatric primitive neuroectodermal tumor biopsies. (**a**) Metaphase fluorescence in situ hybridization (FISH) of a single cell stained with a probe (red) for the chromosome 1 centromere (six copies of chromosome 1 are visible). Scale bar = 100 µm. (**b**) Primary culture of cells from a pediatric primitive neuroectodermal tumor biopsy characterized by a neurosphere-like structure with differentiating cells. Scale bar = 100 µm. (**c**) Dose–response curves of cell viability for two primitive neuroectodermal tumors (**i**, PNET1) and (**ii**, PNET2) and an astrocytoma (**iii**, ACT1) following treatment with the pan-RAR antagonist AGN194310, the RARα-selective agonist AGN195183, or ATRA. Cells were plated into microtitre wells and cell viability determined by measuring the level of cellular ATP at day 5. Data are means ± SE of triplicate values from 3 (**i**), 2 (**ii**), and 4 (**iii**) independent experiments.

**Table 1 ijms-22-03632-t001:** Over and aberrant expression of RARγ in human cancers.

Malignancy	RARγ Expression	Impact on Disease
Acute myeloid leukemia	RARγ chimeric protein	Not responsive to retinoid-based therapy
Colorectal carcinoma	Overexpression	Resistance to chemotherapeutics
Cholangiocarcinoma	Overexpression	Poor prognosis and drug resistance
Hepatocellular carcinoma	Overexpression	Promotes growth of xenografts
Renal cell carcinoma	Overexpression	May contribute to disease progression

**Table 2 ijms-22-03632-t002:** Antagonism of RARγ potently inhibits colony formation by prostate carcinoma cell lines.

Compound	K_d_, nM			IC_50_, nM		
	RARα	RARβ	RARγ	DU145	LNCaP	PC3
Pan-RAR antagonist (AGN194310)	4.3	5	2	34	16	18
RARγ antagonist (AGN205728)	2400	4248	3	6	3	5
RARβγ antagonist (AGN194431)	300	6	70	88	99	103
RARα antagonist (AGN196996)	3.9	4036	>10 K	201	203	235
Pan-RAR agonist (ATRA)	ND	ND	ND	402	344	419

The K_d_ values for affinity are for binding to baculovirus-expressed RARα, RARβ and RARγ DU145 and PC3 cells express RARα and RARγ and LNCaP cells express RARα, β and γ. IC50 values are the concentration of compound that inhibited plate colony formation by 50% by serum free (ITS^+^)-grown LNCaP, DU145 and PC3 cells. For the pan-RAR and RARγ-selective antagonists, the IC_50_ values are close to the K_d_ values.

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
