# Peer review of "The RARγ Oncogene: An Achilles Heel for Some Cancers"

_ijms, 2021, doi:10.3390/ijms22073632_

Round 1
Reviewer 1 Report
This review focuses on cancer stem cells and the role of retinoic acid signaling as an oncogene that may target CSCs. The topic is of interest; however, this review requires extensive revision for publication. In particular, the review needs to focus on the topic (RAR's role as targetable oncogene in CSCs) much earlier. For example, the term Achilles heel is used in the title, but this does not appear in the body of the review until line 348 of 392. Generally, the review is too unfocused and meandering.
Specific comments:
- The authors should consider revising the title. I suggest something along the lines of: "The RARgamma oncogene: an Achilles heel for some cancers"
- The abstract claims that eradicating CSCs is a "bona fide curative strategy". This is dubious at best. The bulk of the tumor ultimately kills the patient. Eradicating CSCs may be necessary for some cancer, but there is no evidence that it is or ever will be sufficient in most cases.
- Transitions between paragraghs/sections generally need work.
- The authors note that LSCs were not identified in APL. However, APL is then highlighted to illustrate how retinoic acid can be used to target LSCs in APL.
- It's unclear what the abrupt introduction of VEGF has to with targeting CSCS. Actually, given the role of hypoxia in tumorigenesis and potentially in CSC maintenance, there is some evidence that anti-angiogenic factors fuel relapse and resistance, which may be why such efforts were largely abandoned years ago.
- The authors briefly discuss the potential role of epigenetics. A recent study "Retinoid-Sensitive Epigenetic Regulation of the Hoxb Cluster Maintains Normal Hematopoiesis and Inhibits Leukemogenesis" could also be useful in this regard.
- The authors refer to the relationship between HSCs and CFUs. The concept of the LT-CIC assay could be of use here.
- The figures tend to be too specialized for a review.
- The concepts regarding RAR and necroptosis are interesting and should be expanded upon.
Author Response
We would like to thank the reviewer for their constructive comments.
We have mentioned the term Achilles heel early in the manuscript at line 60.
We do not agree with the comment that the review is too “unfocussed and meandering”, particularly as our 2nd reviewer comments “This review is dedicated to the interesting and important problem; the manuscript is well written and illustrated”.
We do not feel that it is important to focus on RAR as a targetable oncoprotein much earlier because there is the need to set the context by first looking at CSCs, then targeted therapies, and then get to RARγ in normal cells before looking at RARγ in cancer cells.
Specific comments:
The authors should consider revising the title. I suggest something along the lines of: "The RARgamma oncogene: an Achilles heel for some cancers".
We thank the reviewer for this suggestion, and have changed the title accordingly.
The abstract claims that eradicating CSCs is a "bona fide curative strategy". This is dubious at best. The bulk of the tumour ultimately kills the patient. Eradicating CSCs may be necessary for some cancer, but there is no evidence that it is or ever will be sufficient in most cases.
Lines 54–60 bring to attention the need to kill the bulk of the tumour cells, and how this might be brought about.
Lines 11 and 12: In the abstract we have added the following “may provide a bona fide curative strategy, though there is also the need to kill the bulk cancer cells”.
Lines 342–350 and lines 360–365: We have also brought to attention that the pan- and γ-selective RAR antagonists kill both CSC-like and non-CSC-like cells.
Transitions between paragraphs/sections generally need work.
As we have not re-ordered the sections, we do not feel that transitions need work.
The authors note that LSCs were not identified in APL. However, APL is then highlighted to illustrate how retinoic acid can be used to target LSCs in APL.
Lines 135–137 Thank you for bringing this matter to attention. However, there is evidence to suggest that APL arises in a hematopoietic stem cells that would then give rise to LSC. We have added this information and referenced.
It's unclear what the abrupt introduction of VEGF has to with targeting CSCS. Actually, given the role of hypoxia in tumorigenesis and potentially in CSC maintenance, there is some evidence that anti-angiogenic factors fuel relapse and resistance, which may be why such efforts were largely abandoned years ago.
We consider solid tumours because there is means to targeting this large group of cancers in general which is to prevent the growth of the new blood vessels that feed the tumour. Whilst not targeting CSCs, this is an important avenue to targeted therapies. The approach of targeting blood vessel growth has not been abandoned years ago because colleagues are working on a program to develop new approaches to preventing blood vessel growth.
Lines 164–167 – Thank you for bringing to attention the importance of hypoxia and we have mentioned in terms of reducing hypoxia to increase tumour drug uptake, and referenced.
The authors briefly discuss the potential role of epigenetics. A recent study "Retinoid-Sensitive Epigenetic Regulation of the Hoxb Cluster Maintains Normal Hematopoiesis and Inhibits Leukemogenesis" could also be useful in this regard.
Lines 267–270. Thank you for bringing the paper to our attention and we have included information and referenced.
The authors refer to the relationship between HSCs and CFUs. The concept of the LT-CIC assay could be of use here.
Lines 76–80 Thank you for bringing to attention and we have added the concept of LT-CIC.
The figures tend to be too specialized for a review.
We do not agree that the figures are too specialised. Figure 1 is important to illustrating the ATRA supply to PCa cells and adjacent normal epithelium and how this affects activation of RARγ versus RARα. Our second reviewer comments that “the manuscript is well written and illustrated”.
The concepts regarding RAR and necroptosis are interesting and should be expanded upon.
Lines 401–411 Thank you for bringing to attention and we have added more information and referenced.
Reviewer 2 Report
Review for IJMS
Authors: Geoffrey Brown & Kevin Petrie
Title: RARγ is an oncogene for and Achilles heel to some cancers
This review is dedicated to the interesting and important problem; the manuscript is well written and illustrated. There is a minor criticism:
1) I would recommend to give the definition of abbreviation RAR in Abstract and key words;
2) It is well known that ALDH is one of the markers of a cancer stem phenotype; this enzyme is causally overexpressed in CSCs and involved in their retinoic acid signaling. While the authors describe RARs in CSCs, one can wonder whether there is a link between RARs, ALDH, cancer stemness and CSC resistance to therapeutics? If such a link exists, it would be nice to tell about it in terms of this review;
3) The authors give rather detailed description of the cytotoxic and anti-CSC effects of ALTRA but say nothing about potential combinations of ALTRA with radiotherapy. They should address to such papers: doi: 10.1007/s12015-013-9467-y. doi: 10.1007/s10549-011-1692-y. doi: 10.1038/srep32332. https://doi.org/10.3390/cancers13051102 It seems to me that the authors should consider the point “RAR – anticancer radiotherapy” in their review.
Author Response
We would like to extend our gratitude to the reviewer for their kind and constructive comments.
This review is dedicated to the interesting and important problem; the manuscript is well written and illustrated. There is a minor criticism:
Thank you for the comments about well written and an important problem.
1) I would recommend to give the definition of abbreviation RAR in Abstract and key words.
We have introduced retinoic acid receptor to the abstract and key word.
2) It is well known that ALDH is one of the markers of a cancer stem phenotype; this enzyme is causally overexpressed in CSCs and involved in their retinoic acid signaling. While the authors describe RARs in CSCs, one can wonder whether there is a link between RARs, ALDH, cancer stemness and CSC resistance to therapeutics? If such a link exists, it would be nice to tell about it in terms of this review.
There is a link between ALDH and RAR signalling – Markham et al, Oncotarget, 2019, 10, 1226-1224. We feel that this is a separate story that doesn’t fit well with the content of our review because to my knowledge we can’t link to RARγ.
3) The authors give rather detailed description of the cytotoxic and anti-CSC effects of ALTRA but say nothing about potential combinations of ALTRA with radiotherapy. They should address to such papers: doi: 10.1007/s12015-013-9467-y. doi: 10.1007/s10549-011-1692-y. doi: 10.1038/srep32332. https://doi.org/10.3390/cancers13051102 It seems to me that the authors should consider the point “RAR – anticancer radiotherapy” in their review.
Lines 149 – 151: Thank you for bringing to attention. We use ATRA as an example of a successful RAR-targeted therapy and now therefore only mention, and reference, at the end of the paragraph the use of ATRA with radiotherapy to treat solid tumours.